# Anti-Herpes Simplex Virus Type 1 Activity of *Rosa damascena* Mill Essential Oil and Floral Water in Retinal Infection In Vitro and In Silico

**DOI:** 10.3390/ijms26157521

**Published:** 2025-08-04

**Authors:** Neli Vilhelmova-Ilieva, Rayna Nenova, Kalin Kalinov, Ana Dobreva, Dimitar Peshev, Ivan Iliev

**Affiliations:** 1The Stephan Angeloff Institute of Microbiology, Bulgarian Academy of Sciences, Acad. G. Bonchev Str., 26, 1113 Sofia, Bulgaria; 2Institute for Roses and Aromatic Plants, Agricultural Academy, 49 Osvobozdhenie Blvd., 6100 Kazanlak, Bulgaria; rayna.nenova@gmail.com (R.N.); dr.kkalinov@gmail.com (K.K.); anadobreva@abv.bg (A.D.); 3Department of Chemical Engineering, University of Chemical Technology and Metallurgy, 8 Kliment Ohridski Blvd., 1576 Sofia, Bulgaria; d.peshev@uctm.edu; 4Institute of Experimental Morphology, Pathology and Anthropology with Museum, Bulgarian Academy of Sciences, Acad. G. Bonchev Str., 25, 1113 Sofia, Bulgaria

**Keywords:** *Rosa damascena* Mill essential oil, *Rosa damascena* Mill floral water, rose products, rose oil ingredients, herpes simplex virus type 1, retinal infection, virucidal activity, viral adsorption, cell protection

## Abstract

Recently, essential rose oils and rose products have gained increasing importance in both the cosmetic and food industries, as well as in the composition of medicinal products. We investigated the in vitro antiviral activity of essential oil and floral water from *Rosa damascena* Mill against herpes simplex virus type 1 (HSV-1) infection in rabbit retinal cells (RRCs). The composition of the main chemical components in the rose essential oil was determined by means of gas chromatographic analysis. The effect on the viral replication cycle was determined using the cytopathic effect (CPE) inhibition assay. The virucidal activity, the effect on the adsorption stage of the virus to the host cell, and the protective effect on healthy cells were evaluated using the endpoint dilution method. The effects were determined as deviation in the viral titer, Δlg, for the treated cells from the one for the untreated viral control. The identified main active components of rose oil are geraniol (28.73%), citronellol (21.50%), nonadecane (13.13%), nerol (5.51%), heneicosane (4.87%), nonadecene (3.93), heptadecane (2.29), farnesol (2.11%), tricosane (1.29%), eicosane (1.01%), and eugenol (0.85%). The results demonstrated that both rose products do not have a significant effect on the virus replication but directly affect the viral particles and reduce the viral titer by Δlg = 3.25 for floral water and by Δlg = 3.0 for essential oil. Significant inhibition of the viral adsorption stage was also observed, leading to a decrease in the viral titers by Δlg = 2.25 for floral water and by Δlg = 2.0 for essential oil. When pretreating healthy cells with rose products, both samples significantly protected them from subsequent infection with HSV-1. This protective effect was more pronounced for the oil (Δlg = 2.5) compared to the one for the floral water (Δlg = 2.0). We used the in silico molecular docking method to gain insight into the mechanism of hindrance of viral adsorption by the main rose oil compounds (geraniol, citronellol, nerol). These components targeted the HSV-1 gD interaction surface with nectin-1 and HVEM (Herpesvirus Entry Mediator) host cell receptors, at N-, C-ends, and N-end, respectively. These findings could provide a structural framework for further development of anti-HSV-1 therapeutics.

## 1. Introduction

One of the most common viral agents of ocular disease is herpes simplex virus type 1 (HSV-1) [1,2,3]. As an enveloped virus, HSV-1 possesses surface glycoproteins [4], which by hiding important epitopes, serve as a covert means of evading the host immune response. Four of the major glycoproteins gB, gD, gH, and gL are essential for the virus attachment and entry into the host cell [5,6,7,8]. HSV-1 is a neurotropic, double-stranded DNA virus that causes a wide variety of diseases. Primary ocular infection with HSV-1 infects the corneal epithelium and may or may not develop into a lytic infection, followed by latent infection in the trigeminal ganglion [9,10,11]. Various factors, such as stress, illness, immunosuppression, or trauma, can lead to reactivation of HSV-1 [12,13]. The frequency and severity of recurrences vary among individuals and can lead to serious conditions, including significant vision loss or blindness [10,13,14]. Ocular herpes infections include a wide range of diseases, including blepharitis, conjunctivitis, uveitis, retinitis, and keratitis [14,15,16]. The infection is most commonly unilateral, but in immunosuppressed patients it can also be bilateral [14,17]. Bilateral disease occurs in 1.3–12% of patients and tends to be more severe [16]. Herpes stromal keratitis (HSK) is the most serious manifestation of ocular herpes infections [18]. Each subsequent recurrence causes further corneal damage through opacity, neovascularization, and scarring [10,16]. The global incidence of HSK is estimated at 1.5 million [19,20]. Ocular herpes infections are also associated with a high rate of neonatal morbidity and mortality [21]. Ocular herpes infections may be caused by HSV-1 alone or may occur in a polymicrobial form with other species [22,23,24].

The main treatment for ocular HSV-1 infection is acyclovir (9-[2-hydroxyethoxymethyl] guanine) (ACV), a purine nucleoside analog that inhibits viral replication [25,26]. ACV can be administered topically, orally, or intravenously [13,16,27]. Numerous analogs of acyclovir have also been developed, such as valacyclovir, famciclovir, ganciclovir, and penciclovir [28,29]. They exhibit different bioavailability. Disadvantages of nucleoside analogs include the occurrence of side effects and the rapid formation of resistant mutants. There is an increased demand for new therapeutics to prevent or alleviate ocular herpes recurrences.

Recently, increasing attention has been paid to the study of various products of natural origin, demonstrating antiviral activity. Due to their natural origin, their side effects are minimized, and resistant mutants are selected relatively slowly and difficultly. One of the plants with a long historical use in the traditional medicine is *Rosa damascena* Mill. It is a perennial shrub, reaching a height of 1–2 m, living up to 50 years. Its use is most often in the perfumery, food, and pharmaceutical industries [30,31,32,33,34]. Several commercial products are obtained from *R. damascena* Mill: rose essential oil, rose water, dried flowers, rose hips, rose concrete, and rose absolute. Rose essential oil, also called liquid gold, is the most valuable rose product. The various industrial methods for its extraction, environmental and geographical conditions, techniques of rose petals collection, and storage before processing can affect the chemical composition of the oil. The main producers of rose essential oil in the world are Bulgaria, Turkey, and Morocco [33]. The yield of rose essential oil is very low (0.3–0.4 mL/kg), which makes it such a valuable product. Various rose products are used in traditional medicine for a variety of medical conditions. They are used as laxatives, diuretics, and tonics, as well as in the treatment of respiratory tract infections, abdominal pain, gallstones, jaundice and all forms of hepatitis, cardiovascular diseases, uterine diseases, and erectile dysfunction [23,31,33,35,36,37,38,39,40]. Rose products are also used to treat depression, sadness, nervous stress, wound healing, mouth ulcers, and skin diseases [30,31,41,42]. Modern research on *R. damascena* Mill has confirmed anticancer, analgesic, anti-inflammatory, anticonvulsant, antifungal, and antiviral activity [31,32,43,44]. Antimicrobial activity has been demonstrated against various bacterial strains [45,46,47,48], aiming at extending the shelf life of foods [31]. Using fruit and vegetable models, the vapor phase of *R. damascena* essential oil has been shown to significantly inhibit microbial growth. With the most significant reductions observed in kiwi and banana models [34]. Rose water is used as an antiseptic for eye washes [31,36], and herbal drops containing *R. damascena* Mill product have shown positive effects in patients with conjunctivitis, dry eye, postoperative cataracts, and other ophthalmological conditions [49,50].

Deciphering the underlying mechanisms of the interactions between viral glycoproteins, envelope proteins, cellular receptors, and natural compounds derived from plants with therapeutic potential will yield novel strategies for antiviral treatment of HSV-1 in ocular infections. In the present study, we investigated the antiviral activity of *R. damascena* Mill essential oil and floral water against different stages of HSV-1 reproduction in a model of infection in rabbit retinal cells. To achieve a deeper understanding of the mechanism of attachment of viral glycoprotein D with the host cell receptors nectin 1 and HVEM, we conducted an in silico molecular docking simulation.

## 2. Results

The chromatographic profile of *R. damascena* Mill essential oil is presented in Figure 1. The norm (ISO 9842:2024) for chromatographic profile is limited to ethanol, phenylethanol, citronellol, nerol, geraniol, methyl eugenol, heptadecane, nonadecene, nonadecane, heneicosane, and a small number of minor components [51]. It is used as a guideline based on many studies of rose essential oils over the years [52].

The chromatographic profile of *R. damascena* Mill essential oil, plotted with retention time (minutes) on the x-axis and picoampere (pA) on the y-axis, showed a series of peaks representing volatile constituents. The peaks’ position on the x-axis indicates the time it took for each compound to elute from the column, while the peak height on the y-axis reflects the concentration of that compound. Data was processed using Agilent chromatography data systems (CDSs) software—OpenLab CDS—Agilent RapidControl.NET. The identification of compounds was performed by comparing retention times and authentic references in a chromatographic system. The quantification of the main compounds was achieved by measuring the peak area, without applying any correction factors. Results were presented as relative percentages in Table 1.

In our study the main active components of the rose oil, therpene alcohols represented 55.74% of the total identified substances as follows: geraniol—28.73% (14.0–22.0%), citronellol—21.50% (20.0–34.0%), and nerol—5.51% (5.0–12.0%). Earlier, Dobreva et al. (2013) demonstrated that oxygenated monoterpenes of the main groups of odor substances in the essential oil of *R. damascena* Mill constitute 70% of all identified components [53]. The results for the composition of the studied *R. damascena* Mill essential oil demonstrate compliance with the ISO 9842:2024 standard. The exceptions are the marginally lower content of Methyl eugenol and the elevated content of the major component—geraniol. Although ISO 9842:2024 provides valuable background information, referring to another study, which concerns enhanced authenticity control of essential oils of fully quantified rose oil compounds from different origins [54]. A large skew of distribution, shifted to the maximum values exceeding the standard values (ISO 9842:2024) was shown for the box of Bulgarian rose oil—geraniol. Our study showed similar higher value quantity exceeding the standard. The box of Bulgarian rose oil methyl eugenol exhibited a skew of distribution shifted to the minimum values, which is similar to the result of our study. Our values appear to fully fit within the limits of the cited study, which proves the authenticity of Bulgarian rose oil.

We also studied some characteristic properties of floral water from *R. damascena* Mill. The results are shown in Table 2. The content of rose oil in rose water was 0.33% (reference value: not less than 0.025%), which indicates that the studied rose water is of good quality. The presence of ethyl alcohol (%) was not detected according to the alcoholometric tables. Results for the essential oil and ethyl alcohol contents are presented in percentages.

This study is focused on the treatment of ocular infection caused by HSV-1. We used the adherent rabbit retinal cell line (RRC) as an in vitro model for ocular infection. It may serve as a model for local, targeted drug delivery platforms to human eyes with HSV-1 ocular infection. For example, Blanco and coworkers developed a rabbit ocular melanoma model by injecting human uveal melanoma cells into the eyes of immunosuppressed rabbits [55]. Another team of Kang S.J. and Grossniklaus created a rabbit model of retinoblastoma to study transscleral drug delivery [56].

To exclude the influence of a potential toxic effect of the tested products during the antiviral studies, the cytotoxicity exhibited by the essential oil and the rose water on RRCs, in which the viral replication takes place, was initially determined. The concentration that affected the cell monolayer by 50% (cytotoxic concentration 50%—CC_50_) and the highest concentration that did not affect the morphology of the monolayer (maximally tolerated concentration—MTC) were determined. For *R. damascena* Mill essential oil at the 48 h of treatment (at which the antiviral experiments are reported), a CC_50_ = 223.8 μg/mL and MTC = 10.0 μg/mL were determined. With the MTC thus determined, most of the antiviral experiments were carried out (influence on extracellular virions, their adsorption to cells, and protective effect on healthy cells). For the same time of treatment, *R. damascena* Mill water did not show significant cytotoxicity and no CC_50_ value could be determined even at 100% concentration. A 50% concentration was adopted as the MTC for conducting the antiviral experiments because this is the highest concentration of rose water at which there is no effect on the cell monolayer. (Figure 2 and Figure 3, Table 3).

The effect of *R. damascena* essential oil and water on HSV-1 replication in RRCs was investigated. The effect on the viral replication cycle was determined using the cytopathic effect (CPE) inhibition assay. Our results showed that none of the tested products reached 50% inhibition of viral replication (Figure 4). Therefore, their 50% inhibitory concentrations (IC_50_) could not be determined and hence their selectivity indices (SIs) could not be determined. The activities of the two rose products were compared to the activity of acyclovir (ACV) (IC_50_ = 1.6 μg/mL; SI = 114.0) (Figure 5).

After proving that *R. damascena* Mill essential oil and water do not have a significant effect on the HSV-1 replication, we decided to investigate the effect of the rose products on the viability of extracellular HSV-1 virions. The study was conducted at different time intervals (15, 30, 60, 90, and 120 min). At the first time interval, both products weakly inhibited the HSV-1 virions, leading to a decrease in the viral titers by Δlg = 1.5. At 30 min of exposure, both products significantly reduced the viral titer by Δlg = 2.0. A dependence of the virucidal effect on the treatment time was clearly observed. The inhibition increased with increasing the exposure time. At 60 min of treatment, the effect was further enhanced, with *R. damascena* Mill water being slightly stronger (Δlg = 3.25). The effect was maintained until the last time interval of 120 min. For *R. damascena* Mill essential oil, a decrease in the viral titer reached Δlg = 3.0 at the 60 min of treatment and also remained up to 120 min after exposure (Table 4).

After establishing that both studied rose products showed virucidal activity against HSV-1 virions, we decided to investigate the influence of the studied samples on the adsorption of HSV-1 to sensitive RRCs. The study was conducted at different time intervals during the viral adsorption (15, 30, 45, and 60 min). In parallel with the experimental samples for each time interval, viral controls containing 10^4^ cell culture infectious doses of 50% (CCID_50_) of HSV-1, untreated with the rose products, were prepared. The titer of the corresponding viral control for each time interval was compared with the titer of the sample treated with the rose product. The corresponding Δlg were determined from the difference between the two titers. At the first time interval, the effect was weak, with *R. damascena* Mill water being slightly stronger (Δlg = 1.5). At 30 min of treatment, a decrease of Δlg = 1.5 in the viral titer was observed for both products. After 45 min, the effect became significant, with *R. damascena* Mill water again being more pronounced (Δlg = 2.0). At 60 min of exposure, *R. damascena* Mill essential oil also reduced the viral titer by Δlg = 2.0. In the case of *R. damascena*, Mill water the effect led to the determination of Δlg = 2.25 (Table 5).

The next stage of our study aimed to reveal whether rose products can have a protective effect on healthy RRCs before they are exposed to viral infection. Time intervals (15, 30, 60, 90, and 120 min) of treatment with different durations before the start of viral infection were tested. At the first time interval, the effect was insignificant. At 30 min, the protection was weak. At 60 min of exposure, *R. damascena* Mill essential oil significantly protected the cells, which leads to a decrease of Δlg = 1.75 in the viral titer. After 90 min of treatment, both rose products demonstrated a distinct protective effect, with *R. damascena* Mill essential oil being slightly stronger (Δlg = 2.0). At the last studied time interval of 120 min, the effect was the strongest, and with *R. damascena* Mill essential oil it was again more pronounced (Δlg = 2.5) (Table 6). A trend for increasing the protective effect with increasing time of exposure to the rose products is obvious.

Analysis of each binding pose was conducted for binding score and presence of molecular interactions between geraniol, citronellol, and nerol targeting gD HSV-1 at N-, C-terminal extensions, and at N-terminal extension at binding site of nectin-1 and HVEM host cell receptors, respectively. The ligands (i.e., geraniol, citronellol, and nerol) were subjected to molecular docking with all target regions of gD molecule using AutoDock Vina (v. 4.2.6). The dock preparation function of MGLTools (v 1.5.7) was utilized to prepare each protein for docking that included hydrogens and Gasteiger charges. Forcefield calculations were applied to analyze the ligand for hydrogen atoms, applicable charges, and rotatable bonds. The grid area for gD area of interactions with host cells receptors nectin-1 and HVEM was defined for the targeting ligands. The simulation of docking was conducted with Genetic Algorithm by binding energy minimization and evaluation with RMSD cluster analysis using the ligand atoms only. The number of conformations was fixed at 10. The results generated 10 binding poses of the ligands studied for the designed target gD. The top-ranked poses of each ligand with low binding energies were found to be within the three definite binding sites of both receptors (nectin-1 and HVEM). We have chosen the lowest energy of binding complexes from 10 conformation possibilities of docking with gD HSV-1 for each ligand. The main interactions were hydrophobic and through hydrogen bonding. The studied compounds of rose oil, which are terpene alcohols, indicate that only the hydroxyl groups are indispensable for hindering viral adsorption to the host cell receptors nectin-1 and HVEM. The lowest binding energies of geraniol for all specifics interaction surfaces of gD HSV-1, with the three main rose oil compounds (ligands) are shown in Table 7. The interactions for the geraniol, citronellol, and nerol targeting the gD HSV-1 at nectin-1 and HVEM cellular receptors are presented in Figure 6, Figure 7, and Figure 8, respectively.

## 3. Discussion

In previous studies it has also been reported that citronellol, geraniol, nonadecane, and nerol are main components in the Bulgarian rose oil [45,57]. The literature data reveal slight variability in the chemical composition of Bulgarian rose oils [58]. Moreover, citronellol, geraniol, and nonadecane are proven to be the principal components in all the studied rose oils [51,59,60]. In oil from northern Iran, hexatriacontane and n-tricosane have also been found in high concentrations [61]. Other studies have also indicated nonadecene, phenylethyl alcohol, eugenol, gallic acid, quercetin, and heneicosane as important components of the rose essential oil [59,60].

Katsukawa M. et al. (2011) [62] assessed the impact of rose oil on the peroxisome proliferator-activated receptor (PPAR) and cyclooxygenase-2 (COX-2) in an in vitro experiment. Authors showed that citronellol and geraniol, the major components of rose oil, activated PPARα and γ, and suppressed LPS-induced COX-2 expression in cell culture assays, although the PPARγ-dependent suppression of COX-2 promoter activity was evident only with citronellol. This indicated that citronellol and geraniol were the active components of rose oil.

Jelyani et al., 2021 conducted a study for the food industry that attempted to distinguish the differences between the two rose water samples, one original and one artificial, by identifying the chemical volatile compounds present in the original and artificial rose water using GC and GC-MS [63]. Their studies showed that the major rose water volatile components of all the samples, except the artificial water, were phenylethyl alcohol (36.2–57.4%), citronellol (11.86–35%), and geraniol (4.7–55.25%). The concentrations of the main and minor volatiles extracted from the artificial rose water sample varied significantly. Additionally, in the artificial water sample, the phenylethyl alcohol and citronellol had the lowest quantity and geraniol had the highest amount among all samples. Furthermore, the citronellol/geraniol ratio was the lowest, and substances including terpene 4-ol, terpineol, eugenol, methyl eugenol, ethanol, and methanol were not found in the artificial samples. Two of the original samples had the highest citronellol/geraniol ratios, at 5.3% and 2.87, respectively [63]. Other authors reported that the main components of rose water are geraniol, citronellol, phenylethyl alcohol, and nerol, as well as heneicosane, nonadecane, tricosane, citronellal, geraniol, and citronellol [46,61,64].

In our previous in vitro experiments [65] on the usage of the rose oil in biological studies, we created an experimental setup with human tumorogenic and non-tumorogenic cell cultures; 50% cytotoxic concentrations of rose oil were determined. Our results showed that the studied rose oil is not cytotoxic and has a high level of photosafety—a mandatory requirement for substances that are candidates for therapeutic agents in the treatment of tumor diseases (WHO and US Food and Drug Administration—FDA regulations). CC_50_ served also as a basis for determining the initial dosage in in vivo toxicological experiments with experimental animals. Our findings highlighted the potential of Bulgarian rose oil as a candidate for complementary therapy for colorectal cancer.

In addition to the many biological activities mentioned earlier, rose oils also act as antivirals against various viruses [66,67]. The pharmacological activities of the oil, including antiviral activity, are due to the macrocomponents contained in rose oil. Geraniol, as the substance in the highest concentration in rose oil, has shown antiviral activity in a number of studies [35]. Geraniol showed an inhibitory effect on angiotensin-converting enzyme 2 (UniProtKB-Q9BYF1), SARS-CoV-2 main protease (PDB-6LU7), and SARS-CoV-2 spike glycoprotein [68,69]. It is suggested that inhibition is achieved by topographical complementarity between geraniol and the target viral molecules [69]. When geraniol, citronellol, and limonene were tested for their antiviral activity, all three substances downregulated the ACE2 receptor as well as the protein expression and the mRNA level in HT-29 cells [70].

A number of other major components in rose oil also demonstrate antiviral activity. Citronellol and eugenol exhibit the greatest activity against the influenza virus after direct exposure for only 10 min. Nerol, citral, citronellal, citronellol, geraniol, and eugenol exhibit inhibitory activity against herpes simplex virus type-1 [71]. Citral and nerol demonstrated anti-DENV activity against serotypes 1–4 in vitro established by measuring the reduction in viral NS1 and cell-surface E proteins in HepG2 and Vero cells [72]. Citral also reduced the viral infectivity when the virus was treated before and after entry into the host cells [73]. The antiviral activity of citral has also been studied in vitro, where it has been shown to block the replication of the yellow fever virus [74]. Citral also inhibits infection with non-enveloped murine norovirus (MNV). The effect is time-dependent and is most likely due to nonspecific interaction with the surface of the virions, which prevents the attachment of the virus to sensitive cells.

The present study is focused on the treatment of ocular infections caused by HSV-1. Another research team has also focused its attention in this direction. They demonstrated a slowdown in the growth of herpesvirus-induced keratitis in mouse models under the action of eugenol [75]. Direct inactivation of HSV particles by eugenol has also been described [76].

In our previous study (data not yet published) we investigated the anti-herpes viral activity of geraniol, citronellol, and nerol. The study was conducted with Victoria strain (HSV-1) replicated in MDBK cells. All three components showed an effect on the intracellular replicative cycle, demonstrating SI = 12.0 (for geraniol), SI = 8.5 (for citronellol), and SI = 6.2 (for nerol). All three substances also directly inhibited the viral particles. The effect was time-dependent, with increasing exposure time increasing the inhibitory effect. At 120 min of exposure, citronellol showed the strongest effect, leading to a decrease in the viral titers by Δlg = 3.25. For nerol and geraniol, the effect was Δlg = 2.75. The study also proved that all three components contained in rose oil have an inhibitory effect on the viral adsorption stage, achieving a decrease in the viral titer by Δlg = 2.5 for each of them separately. From the results presented by various research teams, it can be concluded that rose essential oils and products can positively influence and alleviate the course of multiple viral infections. Our previous and current studies, in which we investigated the antiviral activity of essential oils from *R. damascena*, *R. alba*, *R. centifolia*, and *R. gallica* (unpublished data), as well as various rose products, such as wastewater from rose oil production, and rose concrete and absolute, are in line with the observed effects [44,77].

Several publications have examined the primary involvement of nectin-1 and HVEM host cell receptors in HSV-1 adsorption and fusion processes. The process of a viral entry into the host cell is a multi-step process and involves viral envelope glycoproteins gD, gH/gL and gB of HSV-1, which is the primary fusogen directly responsible for merging the viral particle with the host cell membrane. HSV-1 gD/nectin-1 interaction is crucial for the HSV-1 entry [6]. Viral glicoprotein D undergoes conformational changes: displacement of the C-terminus (nectin-1) and folding of the gD N-terminus in the case of HVEM receptor binding [78]. According to the N- and C-terminal extensions of gD are responsible for nectin-1 recognition. For this reason we have chosen both N- and C-terminal residues of gD for ligands. Detailed information for the atomic interactions can be seen in Figure 4 in the publication of Yue D. et al., 2020 [79], where the nectin-1 receptor and the gD protein are shown in surface and cartoon representations, respectively. The interface components in BHV-1 gD, including the N-loop and the multiple elements in the C-terminal extension, are highlighted and marked with patch numbers 1 and 2. We have chosen for the N-terminal extension the N-loop (patch 1), and for C-terminal extension residues patch 2, which includes G/α2 interloop, α3′, and α3/α3′ interloop. The study proves that the binding interface of bovine and human nectin have the same amino acids sequence [79].

Protein crystallization stabilizes the molecules in their lowest free energy conformations—corresponding to the most thermodynamically favorable and structurally stable states. Although our conformational complexes obtained for gD and the three ligands (the main compounds of rose oil: geraniol, citronellol, and nerol) have not shown low binding energy (Table 7), we observed affinity of the ligands for binding to the specific contact internal loops, which can be explained with their bury in topologically comparable surface areas (3D images of Figure 6, Figure 7 and Figure 8). This confirms the mechanism of the proposed model for hindrance of the viral adsorption to the host cell receptors. The later model was concluded based on our in vitro experimental results. We confirmed this with our in silico docking simulation results.

This study has certain limitations. First there is an annual change in the composition of rose oil depending on the climatic conditions during the year (amount of rain, temperature, light, etc.). The period of collection and storage of rose petals is also important. During the storage, large losses and deterioration of the quality of the rose oil can occur. Our study was conducted only in vitro and in silico. Although we have repeatability of the results with our previous studies, it is necessary to continue the study in vivo. Despite the reported docking analysis in this study for the influence of the most active ingredients in the oil with gD HSV-1 at nectin-1 and HVEM biding sites, further studies are needed. In this study the rose oil did not significantly inhibit the viral replication. However, in our previous studies geraniol, citronellol, and nerol showed such activity. Therefore, additional docking analyses will be needed to establish the virus-specific structures which interact with these key rose oil components in the cell.

## 4. Materials and Methods

### 4.1. Host Cell Lines

The adherent cell line of rabbit retinal cells (RRCs) (from the collection of the Institute of Experimental Morphology, Pathology and Anthropology with Museum, Bulgarian Academy of Sciences, Sofia, Bulgaria) was cultured in DMEM/Nutrient Mixture F12 Ham medium (4.5 g/L glucose), 10% fetal calf serum, and 100 IU/mL penicillin and 0.1 mg/mL streptomycin in 25 cm^2^ and 75 cm^2^ plastic cell culture dishes. Cells were maintained in log phase growth at 37 °C and a 5% CO_2_ atmosphere. For in vitro experiments, cells in exponential growth phase after trypsinization were brought to the required concentration and seeded in 96-well cell culture plates. After 24 h of cultivation, under the mentioned conditions, the cells were treated with the test substances, according to the specific experimental setup.

### 4.2. Viruses

Human herpes simplex virus type 1, strain Victoria (HSV-1) was obtained from the National Center for Infectious and Parasitic Diseases (Sofia, Bulgaria). Virus was replicated in a confluent monolayer of RRCs supported with maintenance solution DMEM/Nutrient Mixture F12 Ham medium (4.5 g/L glucose), supplemented with 0.5% fetal bovine serum (Gibco BRL, Scotland, UK) and antibiotics (100 IU/mL penicillin, 100 μg/mL streptomycin). After incubation at 37 °C in a 5% CO_2_ incubator, the virus yield was stored at −80 °C. The infectious titer of the resulting viral stock was determined to be 10^7.0^ CCID_50_/mL.

### 4.3. Reference Compound

Acyclovir {ACV, [9-(2-hydroxyethoxymethyl)-guanine]} was kindly provided by the Deutsches Kresforschung Zentrum, Heidelberg, Germany, with a stock concentration of 3 mM solution in DMSO. Then, falling dilutions were made in DMEM medium to the required concentration.

### 4.4. Candidates for Therapeutic Substances for Complementary Therapy of Retinal Infection with HSV-1–Rosa damascena Mill Essential Oil and Rose Water

Plant material: To ensure the origin and the quality of the rose oil and rose water, flowers of the oil-bearing rose—*R. damascena* Mill were collected from the experimental field of the Institute of Roses, Aromatic and Medicinal Plants, Agricultural Academy, Kazanlak, Bulgaria. The rose picking was carried out during flowering period in May–June 2024, early in the morning between 04:00 and 08:00 a.m. in IV–V phase of development semi-opened to fully opened [80]. The fresh flowers were processed immediately, using the water–steam microdistillation method in Clevinger apparatus to extract rose oil [81,82]. The rose water was produced by the water distillation method in a laboratory still with volume 10 L equipped with a cooler, without an oil separator. Briefly, 1 kg of rose petals was added to 4 L water. The quantity of the obtained rose distillate was 0.7 L.

### 4.5. Gas Chromatographic (GC) Analysis

The GC analysis of the essential oil was conducted using an Agilent 7890A chromatograph (Agilent Technologies, Inc., Santa Clara, CA, USA) equipped with a flame ionization detector (FID). A non-polar capillary column EconoCapTM ECTM-5 (30 m × 0.32 mm × 0.25 µm film of 5% phenyl, 95% methylpolysiloxane) was used. A volume of 0.1 µL of the test sample was injected in split mode 1:10, the inlet temperature was 250 °C, and the FID temperature was set from 60 °C to 300 °C through controlled program. Synthetic air—mixture of N_2_ and O_2_ in ratio (80:20%, respectively) was used as a carrier gas and Hydrogen (99.999% purity) for the FID. To ensure the reliability of analysis, quality control samples of n-alkanes C8–C27 (for the low, medium, and high ranges) in hexane (≥98% purity, Honeywell, Riedel-de Haën TM) were included in the run. The quantification of the main compounds was carried out by peak area without correcting factor. Data was processed using the software OpenLab (Agilent Technologies, OpenLAB CDS version 9.2). The major and some minor constituents of rose oil, identified through GC analysis, are regulated by the international standard ISO 9842:2024 (Figure 1) [83].

### 4.6. Determining the Essential Oil Content in Rose Water

The method involves triple solvent extraction with diethyl ether, dehydration with anhydrous sodium sulfate, solvent removereral via rotary evaporation, and gravimetric determination.

The methodology explicitly references the Standard ISO 9842:2024 and ensures reproducibility. The essential oil content should not be less than 0.025%.

Step 1: Sample preparation

Weigh Rose water into an Erlenmeyer flask.

Step 2: Triple solvent extraction with diethyl ether

Transfer to separating funnel;

Add diethyl ether;

Stir vigorously for 3–5 min;

Collect the lower layer;

Repeat extraction 3 times by adding diethyl ether in decreasing quantity;

Collect the ether layer and transfer into an iodine flask.

Step 3: Dehydration with anhydrous sodium sulfate

Iodine flask containing 5% anhydrous sodium sulfate with collected ether layers shake and dry for 1 h at room temperature.

Step 4: Filtration

Filter through pleated paper filter with sodium sulfate.

Step 5: Solvent remover via rotary evaporation

Transfer filtrate to a distillation flask;

Use rotary vacuum evaporator at 40–47 °C to remove ether;

Dry at 35 °C for 50 min.

Step 6: Gravimetric quantification

Weigh the empty flask (before adding a filtrate for rotary evaporation) and the flask after drying;

Calculate essential oil content in the sample of rose water using the formula:X = (A_2_ − A_1_) × 100/B, %

A_2_—the mass of the flask after drying (Step 5) (g);

A_1_—the mass of the empty flask (g);

B—the mass of the rose water sample (g).

### 4.7. Determining the Ethyl Alcohol Content in Rose Water

In a 250 mL glass boiling flask, 75 mL of rose water and a few grains of pumice were added. The flask was connected to the 200 mm Alin cooler with a ground joint. After boiling, the mixture was collected in a 50 mL measuring tube. The content of ethyl alcohol in rose water was calculated through density determination of liquids by pycnometer—a glass flask with a close-fitting ground glass stopper with a capillary hole through it. This fine hole releases a spare liquid after closing a top-filled pycnometer and allows for obtaining a given volume of measured and/or working liquid with a high accuracy. The empty pycnometer (10 mL volume) was weighed, the sample was transferred into it and was weighed again. The following formula was used.d = (measured sample weight − weight of empty pycnometer)/(weight of distilled water pycnometer − weight of the empty pycnometer) × ρH_2_O (250C) × k (0.0012)

For the calculation of the final result, alcoholometric tables with the relationship between the density of an aqueous alcoholic solution and the content of anhydrous alcohol in the solution were used. The results were presented by volume in percentages.

### 4.8. Cytotoxicity Assay

A confluent monolayer cell culture in 96-well plates (Costar^®^, Corning Inc., Kennebunk, ME, USA) was treated with 0.1 mL/well containing support medium that did not contain/or contained decreasing concentrations of the tested sample. Cells are incubated under the characteristic conditions under which subsequent virus experiments will be performed after 2 days at 37 °C and 5% CO_2_. After the given period of time, the tested products were removed, the cells were washed and incubated with neutral red (NR) dye at 37 °C for 3 h. The 50% cytotoxic concentration (CC_50_) was defined as the concentration of the test sample that reduced cell viability by 50% compared to untreated controls [84]. Each sample was tested in triplicate with four wells per replicate (*n* = 4). The maximally tolerated concentration (MTC) of the tested samples, at which they do not affect the morphology of the cell monolayer, was also determined.

### 4.9. Determination of Infectious Viral Titers

In 96-well plates, a monolayer of RRCs was infected with 0.1 mL of virus suspension at tenfold decreasing dilutions [85]. After the virus adsorption period, the unabsorbed virus was removed and the DMEM medium was added. This was followed by incubation at 37 °C and 5% CO_2_ in a HERA cell 150 CO_2_ incubator (Radobio Scientific Co., Ltd., Shanghai, China) for 48 h. Cells infected with the maximum concentration of the virus and demonstrating the maximum cytopathic effect were used as controls. The resulting cytopathic effect (CPE) was monitored by microscopic observation of the cell monolayer and confirmed by neutral red uptake assay [84].

### 4.10. Antiviral Activity Assay

The cytopathic effect inhibition (CPE) test was used to determine the antiviral activity of rose products [85]. A confluent cell monolayer in 96-well plates was infected with 100 cell culture infectious doses of 50% (CCID_50_) in 0.1 mL. After 1 h of adsorption at 37 °C unattached virus was removed, the tested sample was added at different concentrations and the cells were incubated for 2 days at 37 °C and in the presence of 5% CO_2_. The cytopathic effect was determined using a neutral red uptake assay (incubation with neutral red (NR) dye at 37 °C for 3 h) and the percentage of CPE. Inhibition for each test sample concentration was calculated using the following formula:% CPE = [OD test sample − OD virus control]/[OD toxicity control − OD virus control] × 100
where OD test sample is the mean of the ODs of the wells inoculated with virus and treated with the test sample at the corresponding concentration, ODs virus control is the mean of the ODs of the virus control wells (no compound in the medium), and OD control for toxicity is the mean of the ODs of the wells not inoculated with virus but treated with the corresponding concentration of the test compound. The 50% inhibitory concentration (IC_50_) is defined as the concentration of the test substance that inhibits 50% of viral replication compared to the viral control. The selectivity index (SI) is calculated from the CC_50_/IC_50_ ratio. Each sample was tested in triplicate with four wells per replicate (*n* = 4).

### 4.11. Virucidal Assay

Samples with a total volume of 1 mL containing virus (10^5^ CCID_50_) and tested sample at its maximally tolerated concentration (MTC) in a 1:1 ratio were prepared. A sample containing untreated virus diluted 1:1 with DMEM medium was incubated in parallel. The control and experimental samples were incubated at room temperature for different time intervals (15, 30, 60, 90, and 120) min. Then, by the endpoint dilution method of Reed and Muench (1938) [86], the content of residual infectious virus in each sample and Δlgs compared to untreated controls were determined. Each sample was prepared in quadruplicate (*n* = 4).

### 4.12. Effect on Viral Adsorption

Twenty-four well plates containing a monolayer of RRCs were pre-chilled to 4 °C and inoculated with 10^4^ CCID_50_ of HSV-1. Along with the virus, the monolayer was also treated with the tested samples in their MTC and incubated at 4 °C for the time of virus adsorption. At different time intervals (15, 30, 45, and 60), the virus and the tested sample were removed, the cells were washed with PBS, then the cells were covered with maintenance medium. Incubation followed at 37 °C in the presence of 5% CO_2_ for 24 h. After freezing and thawing three times, the infectious viral titer of each sample was determined. It was compared to the viral titer of the untreated viral control for the given time interval, and Δlgs were determined. Each sample was prepared in quadruplicate (*n* = 4).

### 4.13. Pre-Treatment of Healthy Cells

A monolayer of RRCs previously grown in 24-well cell culture plates (CELLSTAR, Greiner Bio-One GmbH, Kremsmünster, Austria) were treated with the tested sample in their MTC. The samples were incubated for different time intervals of 15, 30, 60, 90, and 120 min at 37 °C. After the given time interval, the samples were removed and the cells were washed with PBS and inoculated with the virus (10^4^ CCID_50_ in 1 mL/well). After 60 min of adsorption, unadsorbed virus was removed and the cells were covered with support medium. Samples were incubated at 37 °C and 5% CO_2_ for 24 h. This was followed by triplicate freezing and thawing of samples and determination of infectious virus titers. Δlg compared to the viral titer of the control (untreated with sample) for the given time interval was determined. Each sample for each time interval studied was prepared in four replicates (*n* = 4).

### 4.14. In Silico Molecular Docking Simulations

The 3D crystal structure (3U82) of the herpes simplex virus glycoprotein D binding sites to host cell adhesion receptor nectin-1 were retrieved from the Protein Data Bank (https://www.rcsb.org/structure/3U82 (accessed on 12 July 2025)). The 3D structure (1JMA) of herpes simplex virus glycoprotein D binding to cellular receptor HVEA/HVEM were downloaded from the Protein Data Bank (https://www.rcsb.org/structure/1JMA (accessed on 13 July 2025)). The 3-D structure formulas of geraniol (CID: 637566), citronellol (CID: 8842), nerol (CID 643820), and linalool (CID: 6549) were downloaded from NCBI PubChem (https://pubchem.ncbi.nlm.nih.gov (accessed on 12 July 2025)).

The ligand and the receptor were prepared before docking using the BIOVIA Discovery Studio Visualizer 2025 (v25.1.0.0) and AutoDock Vina (v4.2.6) [87,88], MGLTool (v1.5.7), including by removing water, adding hydrogen, assigning charges, and generating molecular surfaces. Docking simulations were performed using the AutoDock Vina [89]. The docking pose with the lowest binding energy (kcal/mol) was considered as the most suitable. BIOVIA Discovery Studio Visualizer software was used to visualize the ligand and the receptor interaction in 3D and 2D structures, respectively.

### 4.15. Statistical Analysis

The statistical analysis of the results was performed using one-way ANOVA followed by Dunnett’s post hoc test by GraphPad Prism 8 software (San Diego, CA, USA). A *p*-value of less than 0.05 was considered statistically significant. All results are presented as mean ± SD.

## 5. Conclusions

When investigating the anti-herpesvirus activity of *R. damascena* Mill essential oil and rose water in a model of infection in RRCs, a significant effect on the viral adsorption stage was found. A distinct virucidal effect was also reported, as well as significant protection of healthy cells upon pretreatment with the two rose products. In an attempt to elucidate the underlying molecular mechanisms of antiviral activity, we used the in silico molecular docking method. We explained partially the mechanism of hindrance of the viral adsorption of the main rose oil compounds (geraniol, citronellol, nerol) targeting HSV-1 gD interaction surface with nectin-1 and HVEM host cell receptors, at N-, C-ends, and N-end, respectively. These findings provide a structural framework for further development of anti-HSV-1 therapeutics.

## Data Availability

The raw data supporting the conclusions of this article will be made available by the authors on request.

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
