# Peer review of "Anti-Herpes Simplex Virus Type 1 Activity of Rosa damascena Mill Essential Oil and Floral Water in Retinal Infection In Vitro and In Silico"

_ijms, 2025, doi:10.3390/ijms26157521_

Round 1

Reviewer 1 Report

Comments and Suggestions for Authors

Author Response

Thank you for your thorough review of our manuscript, “ Anti-Herpes Simplex Virus Type 1 Activity of Rosa damascena Mill Essential Oil and Floral Water in Retinal Infection in Vitro and in Silico ”We appreciate your insightful comments and suggestions, which have greatly helped us improve the quality and clarity of our work.

We have carefully considered each issue raised and have made the following edits:

Comments 1: Please include information on whether permits or permissions were obtained for collecting Rosa damascene plant material from the Institute’s experimental fields, in compliance with relevant regulations.

Response 1: Thank you for raising this important point. We confirm that all plant material (Rosa damascena Mill) used in this study was collected from the experimental fields of the Institute of Roses Aromatic and Medicinal Plants (Институт по розата и етеричномаслените култури - IRAMP), Kazanlak, Bulgaria in compliance with relevant national regulations. IRAMP holds both the ownership and the intellectual property rights to the plant material cultivated within its experimental fields. See “Official variety list of varieties if tobacco and vine, fruit medical and aromatic plant species accepted for certification and marketing on the territory of the Republic of Bulgaria 2025”, linked at: https://iasas.government.bg/att/OSL%202%20-%202025%20May%201.pdf.

We have included this in the text at row 356.

Comments 2: The method for determining essential oil content in floral water is buried in the text. A schematic or flowchart summarizing the extraction and quantification steps would aid clarity and accessibility.

Response 2: Thank you for your valuable comment. We appreciate your suggestion to enhance the clarity of the method for determining essential oil content in rose water. In response, we have added a schematic (Scheme 1) summarizing the quantification steps. It has been replaced in section “Materials and Methods”, subsection 4.6 with the following text accompanied with the scheme.

Comments 3: Clearly list all chemical standards used in the GC analysis (e.g., geraniol, nerol standards) to facilitate reproducibility.

Response 3: Thank you for your valuable comment. We have provided a comprehensive list of all pure chemical standards used to identify compounds by matching retention times in our GC analysis. It is included in the supplementary file as Table S1.

Comments 4: Table 1 deviates from the ISO 9842:2024 standard (e.g., elevated geraniol, low methyl eugenol). These deviations and their implications for biological activity or standardization should be addressed. Also correct the typo in the table.

Response 4: We thank the reviewer for highlighting the deviations from ISO 9842:2024 and the typo in Table 1.

While ISO 9842:2024 provides a valuable baseline, referring to the study of Raeber and Steuer, 2023, which concerns enhanced authenticity control of essential oils, we pay attention to box plots (fig. 2) of fully quantified rose oil compounds from different origins. A large skew of distribution, shifted to the maximum values exceeding the standard values (ISO 9842:2024) was shown for the box of the Bulgarian rose oil - geraniol. Our study showed similar higher value quantity exceeding the standard.  The box of Bulgarian rose oil methyl eugenol exhibited skew of distribution shifted to the minimum values which is similar to the result of our study. Our values appear to fully fit within the limits of the cited study, which proves the authenticity of Bulgarian rose oil.

Comments 5: The manuscript does not adequately correlate the phytochemical composition (Table 1) with observed biological effects. Please include a discussion or table linking each major constituent to known antiviral properties from literature and, where applicable, experimental data.

Response 5: In the Discussion section of the manuscript, lines 380 to 298 and 395 - 409 describe the main components contained in rose oil. The data are both our own and the results of other authors. Lines 423 to 445 describe the antiviral activity of the main components of rose oil, described by other authors. Lines 451 to 461 describe our personal results on the antiviral activity of the main components of rose oil, which have not been published to date.

Comments 6: The absence of a negative control in the virucidal assay (Table 4) is a critical limitation. A vehicle-only control should be included to confirm that observed effects are not due to the solvent or dilution effects. Please revise the experimental setup accordingly and update the interpretation of results.

Response 6: In parallel with the prepared samples, a sample is prepared that contains only virus, untreated with a pink product. It is only in the presence of the nutrient medium needed to nourish the cells. No other solvent is contained in the medium. This control is precisely the viral control against which Δlgs were calculated, therefore it is not presented in the table.

Comments 7: The viral adsorption assay (Table 5) lacks a clear explanation of how Δlg values were calculated. A detailed description of the calculation method and formula, including the role of control titers, should be provided to ensure clarity and reproducibility.

Response 7: An explanation has been added to the Results section on how to calculate Δlg values. There is also an explanation in section 4. 12. Effect on viral adsorption.

Comments 8: Table footnotes (e.g., for Tables 4–6) should be expanded to define terms such as Δlg, MTC, and specific assay conditions.

Response 8: The notes below the tables have been expanded.

Comments 9: Claims regarding the protective mechanism involving binding of rose oil components to viral glycoproteins are speculative. Without supporting experimental evidence, such as binding assays or molecular docking, this hypothesis should be either removed or experimentally validated.

Response 9: Our claim was made based on the presented studies of other authors. We now present molecular docking results to support our claim.

Comments 10: Please consider adding in silico molecular docking simulations of major compounds (e.g., geraniol, citronellol, nerol) with HSV-1 glycoproteins (gD, gB) to support the proposed mechanism of viral entry inhibition.

Response 10: Thank you for the suggestion. We used molecular docking to demonstrate the potential binding of the three most active components in rose oil to gD HSV-1 at nectin -1 and HVEM binding sites.

Comments 11: Acyclovir (ACV) should be included in all antiviral assays—not just cytotoxicity testing—for comparison. Its performance across virucidal, adsorption inhibition, and protective assays would provide a critical benchmark for efficacy.

Response 11: The activity of ACV is included as a reference substance in Figures 2 and 3 (cytotoxicity and effect on the viral replicative cycle). In the other antiviral experiments, ACV was not used as a reference substance, because it would not be scientifically justified. ACV is the first discovered nucleoside analogue, which subsequently received the widest application. Its mechanism of action was described decades ago. ACV itself is applied in an inactive form and its activation occurs only inside the infected cell by the virus-specific enzyme Thymidine Kinase (TK). Without the presence of this enzyme, ACV is inactive. This means that outside the infected cell it cannot show any activity. It would have no effect on either the virions or their adsorption to the cell. ACV is a specific nucleoside analogue, without other forms of action.

Comments 12: Essential oil failed to inhibit viral replication significantly at safe concentrations (MTC = 10 µg/mL). Since IC50 values were not reached, this suggests low potency in its current form.

Response 12: The assumption of the esteemed reviewer is correct. Our other studies on other viral strains and cell lines showed the same results. Lack of significant effect (less than 50% inhibition of viral replication) on the replicative cycle of the virus, but significant inhibition of the viral adsorption stage, virucidal activity and protective effect on cells. As I mentioned in the discussion, we have studied some of the components of rose oil in a pure state. They show an effect on viral replication. From here follows the conclusion that the dose and bioavailability of the active components in the cell are important. Since their content in the oil is lower than the pure product itself, hence the activity is lower. Our other studies show that if a substance or product containing a mix of multiple substances (such as rose oil), if it exhibits even weak activity against viral replication (such as rose oil), if it is introduced into a carrier, its activity can be increased many times. Since we have achieved the effect with other products, this is a goal of our future studies, which we expect to increase the currently weakly observed effect.

Comments 13: The statistical analysis is insufficient for the experimental design. Use of only Student’s t-test is inappropriate for multi-group or time-point comparisons. Oneway or two-way ANOVA with suitable post hoc tests (e.g., Dunnett’s or Tukey’s) should be applied to Figures 2 and 3, and Tables 4–6. Report the statistical test used in figure legends, and clearly indicate significance levels. All figures and tables presenting quantitative data should include error bars (SD or SEM), n values (biological replicates), and clear statistical annotations.

Response 13: We thank you for the suggestion to improve and expand the statistical analysis of the conducted experiments. In the revised version of the manuscript we used one-way ANOVA analysis of variance followed by Dunnett’s post hoc test. It was applied to Figures 2 and 3, as well as to Tables 4-6. SD is also included for Tables 4-6.

Comments 14: Figures are of low resolution and lack fully labeled axes and detailed legends. Replace with high-quality versions and include ACV data for direct visual comparison where applicable.

Response 14: The figures have been revised. The data for acyclovir are included for comparison with rose oil. In the figures with rose water it is not included because acyclovir and rose water have different units of concentration and the comparison will not be correct.

 Comments 15: Some units are inconsistently formatted (e.g., µg/mL vs μg / ml). Please standardize unit formatting using SI conventions throughout the manuscript.

Response 15: Thank you for your comment. The entire manuscript has been reviewed and the formatting has been standardized.

 Comments 16: Maintain consistent terminology throughout the manuscript (e.g., use either “retinal cells” or “RRC” uniformly).

Response 16: The entire manuscript has been reviewed and the abbreviation RRC is indicated throughout, except at the beginning, where the abbreviation itself is introduced.

Comments 17: Reference formatting needs improvement. Remove redundant citations and ensure correct placement of references within the text. The discussion at times includes multiple references for similar claims. Consolidate citations to enhance readability and avoid redundancy.

Response 17: Redundant references have been removed, those that have been retained are necessary for the presentation of the information in the manuscript. Where possible, citations have been merged. The numbering of references in the text and in the References section has been corrected.

Comments 18: The manuscript contains numerous lengthy and awkwardly structured sentences. A comprehensive language edit is necessary to improve clarity, grammar, and scientific readability.

Response 18: The entire manuscript has been reviewed and some of the sentences have been restructured.

Comments 19: The discussion section should acknowledge study limitations, including the in vitro nature of the model, variability in rose oil composition due to origin or extraction methods, and the lack of mechanistic evidence.

Response 19: A new, final paragraph has been added to the Discussion section, in which we point out the limitations of this study.

Comments 20: The conclusion currently overstates the findings by suggesting therapeutic efficacy against ocular HSV-1 infections. Please revise this to reflect the preliminary, in vitro nature of the results and the necessity of further in vivo studies.

Response 20: The conclusion has been revised and the data from the new results have been included.

Reviewer 2 Report

Comments and Suggestions for Authors

I have reviewed the manuscript titled “ Anti-Herpes Simplex Virus Type 1 Activity of Rosa damascena Mill Essential Oil and Floral Water in Retinal Infection in Vitro”. The manuscript presents relevant information about the evaluation of the antiviral activity of natural products derived from Rosa damascena (essential oil and floral water) against herpes simplex virus type 1 (HSV-1) in an in vitro rabbit retinal cell model. The topic is relevant in the context of the search for natural therapeutic alternatives in the face of growing resistance to conventional antivirals. The study presents an interesting experimental approach by evaluating three stages of the viral cycle (virucidation, adsorption, and cellular protection), and it provides a chemical analysis of the rose oil. However, the work presents methodological, analytical, and conceptual weaknesses; therefore, the manuscript can not be published in its current form.

Comments:

  • Substantial improvements to the writing of the entire manuscript are recommended. Numerous excessively short paragraphs have been identified that do not fully develop the primary or secondary ideas. Furthermore, there are recurring problems with grammar, punctuation, and style. A professional linguistic review is recommended to align the text with international standards for scientific writing.
  • Standardize the formatting of decimal numerical values: In some cases, a decimal point (.) is used, and in others, a comma (,) is used, which causes confusion.
  • Correct typographical formatting should be applied: scientific names should be italicized (such as Rosa damascena), as should Latin terms such as in vitro.
  • Enhance the aesthetic presentation of tables by ensuring uniform formatting, clear headings, and comprehensive explanatory notes.
  • Review and improve all figures. Complete details should be included on the X and Y axes (e.g., what is being measured or inhibited, units, exact concentrations). If multiple graphs are presented in a single figure, they should be separated into subsections (e.g., Figures 2A and 2B) and clearly explained in the legend. It is recommended to avoid the use of abbreviations within figures so that they can be interpreted independently, without having to refer to the main text.
  • The first time the acronyms CC₅₀ and MTC are mentioned in the results section, they should be defined and explained, as this is only done later in the methodology. This will facilitate the reader's understanding of the initial results.
  • On line 176, the decision to use a 50% rosewater concentration in subsequent analyses should be justified. The choice should be clearly supported.
  • In Table 3, there is an inconsistency between the significance asterisks. In the table, a single asterisk (*) appears, while in the footnote, three appear (***). Furthermore, the statistical significance corresponding to each symbol is not adequately specified. This point should be corrected.
  • In all tables, where applicable, the mean ± standard deviation should be reported, and the number of replicates performed should be clarified.
  • In Tables 4, 5, and 6, and where applicable, it should be explicitly stated that the reported time corresponds to the time of exposure of the virus or cell to the treatment.
  • Although the manuscript provides background on the antiviral activity of some individual components of the essential oil, the possible mechanism of antiviral action needs to be further developed, and concrete hypotheses should be proposed in the discussion.
  • In some sections of the methodology, the number of replicates performed and the specific experimental conditions (e.g., precise concentrations, missing units in some graphs) are unclear. This affects the reproducibility of the study and should be corrected.
  • On line 458, the time units (minutes, hours, etc.) should be added for correct interpretation of the data.
  • All abbreviations should be defined the first time they are mentioned in the text.
  • The use of the cytopathic effect (CPE) assay is mentioned in the abstract and methodology, but this term does not appear in the results section, leading to confusion about which part of the results it refers to. The same is true for IC₅₀ and SI: they are described in the methodology, but no related results are reported or discussed.
  • The conclusion states that "some molecules in the composition of rose oil probably can bind to glycoproteins...", although this hypothesis has not been discussed or supported anywhere in the manuscript. No experimental data or molecular tools (such as in silico modeling) are presented to support this statement, so it should be removed or reformulated as a perspective to be studied in future work.
  • The conclusion also states that the products evaluated "can be used to reduce the course of the disease of herpes eye infection," which is an overly ambitious statement, as no activity on intracellular replication of the virus was demonstrated, nor were in vivo assays performed. This conclusion should be tempered and strictly limited to the effects observed under in vitro conditions.

Author Response

Thank you for your thorough review of our manuscript, “ Anti-Herpes Simplex Virus Type 1 Activity of Rosa damascena Mill Essential Oil and Floral Water in Retinal Infection in Vitro and in Silico ”We appreciate your insightful comments and suggestions, which have greatly helped us improve the quality and clarity of our work.

We have carefully considered each issue raised and have made the following edits:

Comments 1: Substantial improvements to the writing of the entire manuscript are recommended. Numerous excessively short paragraphs have been identified that do not fully develop the primary or secondary ideas. Furthermore, there are recurring problems with grammar, punctuation, and style. A professional linguistic review is recommended to align the text with international standards for scientific writing.

Response 1: Thank you for your comment. The entire manuscript has been revised, short paragraphs have been removed, and the style has been improved.

Comments 2: Standardize the formatting of decimal numerical values: In some cases, a decimal point (.) is used, and in others, a comma (,) is used, which causes confusion.

Response 2: The formatting of decimal numeric values is standardized.

Comments 3: Correct typographical formatting should be applied: scientific names should be italicized (such as Rosa damascena), as should Latin terms such as in vitro.

Response 3: The entire manuscript has been reviewed and omissions have been italicized.

Comments 4: Enhance the aesthetic presentation of tables by ensuring uniform formatting, clear headings, and comprehensive explanatory notes.

Response 4: Thank you for your comment. We have improved the aesthetic presentation of the tables.

Comments 5: Review and improve all figures. Complete details should be included on the X and Y axes (e.g., what is being measured or inhibited, units, exact concentrations). If multiple graphs are presented in a single figure, they should be separated into subsections (e.g., Figures 2A and 2B) and clearly explained in the legend. It is recommended to avoid the use of abbreviations within figures so that they can be interpreted independently, without having to refer to the main text.

Response 5: All figures have been revised. They are divided into subfigures and are clearly explained below the figure. Abbreviations within the figures have been removed.

Comments 6: The first time the acronyms CC₅₀ and MTC are mentioned in the results section, they should be defined and explained, as this is only done later in the methodology. This will facilitate the reader's understanding of the initial results.

Response 6: Thanks for the note. A definition of both abbreviations is provided in the results section.

Comments 7: On line 176, the decision to use a 50% rosewater concentration in subsequent analyses should be justified. The choice should be clearly supported.

Response 7: An explanation for the choice of this concentration has been added to the text.

Comments 8: In Table 3, there is an inconsistency between the significance asterisks. In the table, a single asterisk (*) appears, while in the footnote, three appear (***). Furthermore, the statistical significance corresponding to each symbol is not adequately specified. This point should be corrected.

Response 8: Thank you for pointing out the technical error we made. The appropriate correction has been made.

Comments 9: In all tables, where applicable, the mean ± standard deviation should be reported, and the number of replicates performed should be clarified.

Response 9: The appropriate addition has been made.

Comments 10: In Tables 4, 5, and 6, and where applicable, it should be explicitly stated that the reported time corresponds to the time of exposure of the virus or cell to the treatment.

Response 10: The appropriate addition has been made.

Comments 11: Although the manuscript provides background on the antiviral activity of some individual components of the essential oil, the possible mechanism of antiviral action needs to be further developed, and concrete hypotheses should be proposed in the discussion.

Response 11: Thank you for the suggestion. We conducted an additional dokint analysis to prove the potential binding of the three most active components of rose oil with gD HSV-1 at nectin -1 and HVEM binding sites. As a result of the results obtained, we included a new discussion in the Discussion section.

Comments 12: In some sections of the methodology, the number of replicates performed and the specific experimental conditions (e.g., precise concentrations, missing units in some graphs) are unclear. This affects the reproducibility of the study and should be corrected.

Response 12: Thank you for pointing out our omission. The number of repetitions is indicated everywhere in the methodology. Corrections have also been made to the graphs.

Comments 13: On line 458, the time units (minutes, hours, etc.) should be added for correct interpretation of the data.

Response 13: Due to the misalignment of the lines, it is not clear which line exactly the note is for here. Based on the logic of the paragraph, we assumed that it was about the incubation time with neutral red and added it.

Comments 14: All abbreviations should be defined the first time they are mentioned in the text.

Response 14: Thank you for your comment. The entire manuscript has been reviewed and the missing definitions of abbreviations at first mention have been added.

Comments 15: The use of the cytopathic effect (CPE) assay is mentioned in the abstract and methodology, but this term does not appear in the results section, leading to confusion about which part of the results it refers to. The same is true for IC₅₀ and SI: they are described in the methodology, but no related results are reported or discussed.

Response 15: Relevant information has been added to the Results section.

Comments 16: The conclusion states that "some molecules in the composition of rose oil probably can bind to glycoproteins...", although this hypothesis has not been discussed or supported anywhere in the manuscript. No experimental data or molecular tools (such as in silico modeling) are presented to support this statement, so it should be removed or reformulated as a perspective to be studied in future work.

Response 16: Thank you for your comment. It prompted us to conduct additional docking analysis. From the results of this analysis, we have included additional discussion in the Discussion and Conclusion sections.

Comments 17: The conclusion also states that the products evaluated "can be used to reduce the course of the disease of herpes eye infection," which is an overly ambitious statement, as no activity on intracellular replication of the virus was demonstrated, nor were in vivo assays performed. This conclusion should be tempered and strictly limited to the effects observed under in vitro conditions.

Response 17: The statement in the conclusion has been revised.

Reviewer 3 Report

Comments and Suggestions for Authors

Dear all, 
articles introducing plant medicine are of great interest, due to the increasing resistance of strains to acyclovir. In the work presented here, we investigated the effects of two substances on different stages of HHV-1 replication in vitro. Below are my comments.

  1. The authors performed a qualitative chemical analysis of the extract studied. It would be of great interest to know what the major metabolites are present in the extract in the drug development context. Can authors provide this information based on their data?
  2. The tables in the presented article do not have the formatting as suggested by the journal. Please correct (Tab 1 and 2).
  3. The caption of the figures could be more expanded. The recipient reading the article should know what method was used to perform a given analysis - by looking only at the figure. There is nothing mentioned about statistics under the figures.
  4. The statistical significance marked in the figures is missing? Were all the analyses not statistically significant ? Which statistical method was chosen by you and why?
  5. In the case of studying the effect of subtension on extracellular virions, have you investigated how the incubation period itself affects the decline in viral activity?
  6. Verses 266-268 different text font
  7. Acyclovir - in what trials was it used as a reference drug. Missing from the figures are comparisons to the effectiveness of acyclovir
  8. In the antiviral test, I would also propose a variant of experience - treatment. Because it would have more meaningful applications in the treatment of infection. E.g. HHV-1 infection of cells and then administration of test substances.
  9. The work would be more enriched, for example, by the presence of images from under the microscope with inhibited cytopathic effect
  10. The gold standard in virological testing is cytopathic effect analysis. Were the authors tempted to examine the level of replication quantitatively by qPCR ? perhaps greater differences would be seen with the preparations tested ?

Author Response

Thank you for your thorough review of our manuscript, “ Anti-Herpes Simplex Virus Type 1 Activity of Rosa damascena Mill Essential Oil and Floral Water in Retinal Infection in Vitro and in Silico ”We appreciate your insightful comments and suggestions, which have greatly helped us improve the quality and clarity of our work.

We have carefully considered each issue raised and have made the following edits:

Comments 1: The authors performed a qualitative chemical analysis of the extract studied. It would be of great interest to know what the major metabolites are present in the extract in the drug development context. Can authors provide this information based on their data?

Response 1: In the Discussion section, lines 380 to 367 and 395 - 409, there is a description of the main components contained in rose oil. Both our results and the results of other authors are briefly presented. From lines 423 to 445, there is a description of the antiviral activity of the main components of rose oil, described by other authors. From lines 451 to 461, our personal results on the antiviral activity of the main components of rose oil, which have not been published so far, are described. That is, these are not only the components in the largest quantities, but also the components to which the biological activities of rose oil and in particular the antiviral activity are due. As our own results show, rose oil shows an inhibition of the viral replicative cycle lower than 50%. However, the pure components studied by us show a significantly higher activity. As we have indicated in the text: SI = 12.0 (for geraniol), SI = 8.5 (for citronellol) and SI = 6.2 (for nerol). This proves that the pure substances have greater activity due to their higher concentration, compared to their content in the oil. This suggests the possibility of their development as therapeutics. This of course requires a lot of additional research, including in vivo studies.

Comments 2: The tables in the presented article do not have the formatting as suggested by the journal. Please correct (Tab 1 and 2).

Response 2: The formatting of the tables has been corrected.

Comments 3: The caption of the figures could be more expanded. The recipient reading the article should know what method was used to perform a given analysis - by looking only at the figure. There is nothing mentioned about statistics under the figures.

Response 3: The captions under the figures have been expanded. The statistical method and the significance of the results obtained are given.

Comments 4: The statistical significance marked in the figures is missing? Were all the analyses not statistically significant ? Which statistical method was chosen by you and why?

Response 4: Thank you for your comment. We have included statistics in the figures. Below the figures we have indicated the method and the significance of the results obtained.

Comments 5: In the case of studying the effect of subtension on extracellular virions, have you investigated how the incubation period itself affects the decline in viral activity?

Response 5: In general, all the results we have presented related to tracking some activity for different time intervals show that the effect increases as the exposure time increases. That is, the effect is time-dependent. We have noted this in the Results section.

Comments 6: Verses 266-268 different text font

Response 6: Thanks for the note. The font has been corrected.

Comments 7: Acyclovir - in what trials was it used as a reference drug. Missing from the figures are comparisons to the effectiveness of acyclovir

Response 7: Acyclovir was used as a reference substance in the study of cytotoxicity and the effect on viral replication. Data on its activity are added in Figures 2 and 3.

Comments 8: In the antiviral test, I would also propose a variant of experience - treatment. Because it would have more meaningful applications in the treatment of infection. E.g. HHV-1 infection of cells and then administration of test substances.

Response 8: We have applied this option, but as we have described in the text the inhibition of viral replication is less than 50%. Therefore, IC50 and hence SI of the tested products cannot be determined. The results are presented in Figure 3.

Comments 9: The work would be more enriched, for example, by the presence of images from under the microscope with inhibited cytopathic effect

Response 9: We attach microscopic images. They were working, without the idea of being published. If you think the quality is not good enough, we will remove them again, because new photos cannot be taken. The experiments were conducted more than a year ago. If they are repeated now, it will not be with the same rose oil and water. The studies will be with materials from a new batch, with a partially different composition and some changes in the values presented. So new photos will not show the effect of the rose products studied and presented in the manuscript.

Comments 10: The gold standard in virological testing is cytopathic effect analysis. Were the authors tempted to examine the level of replication quantitatively by qPCR ? perhaps greater differences would be seen with the preparations tested ?

Response 10: At this stage we have only studied the cytopathic effect. But your suggestion is very interesting. Indeed, larger differences in the results obtained can be found using qPCR. We will keep your suggestion in mind in our next studies

Round 2

Reviewer 1 Report

Comments and Suggestions for Authors

The manuscript has been improved but it's hard to follow because of the track changes. Please always upload a clean manuscript with highlighted changes marked with different colors, not just as a track changes file.